# Projections of incident atherosclerotic cardiovascular disease and incident type 2 diabetes across evolving statin treatment guidelines and recommendations: A modelling study

Joseph C. Engeda[1,2]*, Stefan K. Lhachimi[3,4], Wayne D. Rosamond[1], Jennifer L. Lund[1], Thomas C. Keyserling[5], Monika M. Safford[6], Lisandro D. Colantonio[7], Paul Muntner[7], Christy L. Avery[1]*

1 Department of Epidemiology, University of North Carolina at Chapel Hill, Chapel Hill, North Carolina, United States of America, 2 Department of Epidemiology and Biostatistics, University of California, San Francisco, California, United States of America, 3 Research Group Evidence-Based Public Health, Leibniz Institute for Epidemiology and Prevention Research (BIPS), Bremen, Germany, 4 Department for Health Services Research, Institute for Public Health and Nursing, University of Bremen, Bremen, Germany, 5 Division of General Medicine and Clinical Epidemiology, University of North Carolina, Chapel Hill, North Carolina, United States of America, 6 Division of General Internal Medicine, Weill Cornell Medical College, New York, New York, United States of America, 7 Department of Epidemiology, University of Alabama at Birmingham, Birmingham, Alabama, United States of America

* Jengeda5@gmail.com (JCE); christy_avery@unc.edu (CLA)

**Data Availability Statement:** NHANES data are available from https://www.cdc.gov/nchs/nhanes/

## Abstract

### Background

Experimental and observational research has suggested the potential for increased type 2 diabetes (T2D) risk among populations taking statins for the primary prevention of athero-sclerotic cardiovascular disease (ASCVD). However, few studies have directly compared statin-associated benefits and harms or examined heterogeneity by population subgroups or assumed treatment effect. Thus, we compared ASCVD risk reduction and T2D incidence increases across 3 statin treatment guidelines or recommendations among adults without a history of ASCVD or T2D who were eligible for statin treatment initiation.

### Methods and findings

Simulations were conducted using Markov models that integrated data from contemporary population-based studies of non-Hispanic African American and white adults aged 40–75 years with published meta-analyses. Statin treatment eligibility was determined by predicted 10-year ASCVD risk (5%, 7.5%, or 10%). We calculated the number needed to treat (NNT) to prevent one ASCVD event and the number needed to harm (NNH) to incur one incident case of T2D. The likelihood to be helped or harmed (LHH) was calculated as ratio of NNH to NNT. Heterogeneity in statin-associated benefit was examined by sex, age, and statin-associated T2D relative risk (RR) (range: 1.11–1.55). A total of 61,125,042 U.S. adults (58.5%

index.htm. Data from REGARDS may only be accessible after submitting a proposal to the REGARDS study.

**Funding:** JCE T32HL007055 National Heart, Lung, and Blood Institute; T32AG049663 National Institute on Aging. The sponsor did not play any role in the study design, data collection and analysis, decision to publish, or preparation of the manuscript.

**Competing interests:** The authors have declared that no competing interests exist.

**Abbreviations:** ACC, American College of Cardiology; AHA, American Heart Association; ASCOT-LLA, Anglo-Scandinavian Cardiac Outcomes Trial-Lipid Lowering Arm; ASCVD, atherosclerotic cardiovascular disease; CCS, Canadian Cardiovascular Society; CHD, coronary heart disease; EAS, European Atherosclerosis Society; ESC, European Society of Cardiology; HR, hazard ratio; ICD-10, International Statistical Classification of Disease 10th edition; JUPITER, Justification for the Use of statins in Prevention: an Intervention Trial Evaluating Rosuvastatin; LDL-C, low-density lipoprotein cholesterol; LHH, likelihood to be helped or harmed; Lp(a), lipoprotein(a); NHANES, National Health and Nutrition Examination Survey; NICE, National Institute for Health and Care Excellence; NNH, number needed to harm; NNT, number needed to treat; PSA, probabilistic sensitivity analysis; RCT, randomized controlled trial; REGARDS, Reasons for Geographic and Racial Differences in Stroke; RR, relative risk; STROBE, Strengthening the Reporting of Observational Studies in Epidemiology; T2D, type 2 diabetes; USPSTF, U.S. Preventive Services Task Force.

female; 89.4% white; mean age = 54.7 years) composed our primary prevention population, among whom 13–28 million adults were eligible for statin initiation. Overall, the number of ASCVD events prevented was at least twice as large as the number of incident cases of T2D incurred (LHH range: 2.26–2.90). However, the number of T2D cases incurred surpassed the number of ASCVD events prevented when higher statin-associated T2D RRs were assumed (LHH range: 0.72–0.94). In addition, females (LHH range: 1.74–2.40) and adults aged 40–50 years (LHH range: 1.00–1.14) received lower absolute benefits of statin treatment compared with males (LHH range: 2.55–3.00) and adults aged 70–75 years (LHH range: 3.95–3.96). Projected differences in LHH by age and sex became more pronounced as statin-associated T2D RR increased, with a majority of scenarios projecting LHHs < 1 for females and adults aged 40–50 years. This study's primary limitation was uncertainty in estimates of statin-associated T2D risk, highlighting areas in which additional clinical and public health research is needed.

## Conclusions

Our projections suggest that females and younger adult populations shoulder the highest relative burden of statin-associated T2D risk.

## Author summary

### Why was this study done?

- Statins are a widely prescribed medication used to prevent atherosclerotic cardiovascular disease (ASCVD).

- Past research has suggested the potential for increased type 2 diabetes (T2D) risk among populations taking statins.

- Few studies have compared statin-associated benefits (ASCVD prevention) and statin-associated harms (increased T2D incidence) using different statin treatment guidelines and recommendations in contemporary U.S. populations or by population subgroups defined by age and sex.

### What did the researchers do and find?

- We developed a simulation model to compare statin treatment guidelines and recommendations in a U.S. population without T2D or a previous ASCVD event by synthesizing data from contemporary population-based studies with published meta-analyses.

- Using the model, our projections suggested that the number of ASCVD events prevented was at least twice as large as the number of incident cases of T2D incurred. However, the number of T2D cases incurred surpassed the number of ASCVD events prevented when higher statin-associated T2D relative risks (RRs) were assumed.

- Our models also suggest that females and the youngest adults received lower absolute benefits of statin treatment compared with males and the oldest adults. Projected differences by age and sex also were more pronounced as the effect of statins on T2D was

increased. For females and younger populations, a majority of these scenarios suggested that the number of new T2D cases was greater than the number of ASCVD events prevented.

### What do these findings mean?

- Our projections suggested that females and younger adult populations shoulder the highest relative burden of statin-associated T2D risk.

- Clinical and public health research to more precisely quantify statin's effects on T2D is needed given the sensitivity of projections to this simulation parameter.

## Introduction

Statins are a widely prescribed class of lipid-lowering medication used to prevent atherosclerotic cardiovascular disease (ASCVD) [1, 2]. In 2013, the American College of Cardiology (ACC)/American Heart Association (AHA) updated previous cholesterol treatment guidelines, particularly with respect to ASCVD primary prevention [3]; these guidelines were further revised in 2018 [4]. Previous guidelines emphasized low-density lipoprotein cholesterol (LDL-C) levels for guiding statin treatment [5], while the 2013 ACC/AHA guidelines based statin treatment recommendations on predicted 10-year ASCVD risk. As a result, an estimated 10.4 million U.S. adults were newly eligible for statin treatment for the primary prevention of ASCVD, with adults 60–75 years of age versus other age groups being more likely to be newly eligible [6]. While not formal guidelines, other recommendations have the potential to further expand the population eligible for statin treatment for primary prevention of ASCVD, e.g., expanding guidelines to include populations with predicted 10-year risks of ASCVD of 5% or greater [7, 8].

   Experimental and observational research has suggested the potential for adverse effects of statins, including type 2 diabetes (T2D), a side effect of particular interest because of its associated adverse health outcomes and impact on quality of life [9–13]. Accumulating evidence has suggested that statins increase the relative risk (RR) of T2D by 5%–55% [10–15], with potentially elevated statin-associated T2D risk in females [16], younger populations compared to older populations, and populations with lower LDL-C compared to populations with higher LDL-C [15]. Such findings merit further investigation in light of evolving statin recommendations that target growing proportions of populations for ASCVD primary prevention, for whom the net effects of statins remain incompletely quantified [17]. Therefore, this study projected the number of expected ASCVD events prevented and incident cases of T2D incurred in primary prevention populations across three 10-year ASCVD risk-based statin treatment guidelines or recommendations [6, 18, 19].

## Methods

### Motivation for simulation model

This study has been performed according to the Strengthening the Reporting of Observational Studies in Epidemiology (STROBE) checklist and employed a simulation model to examine intended and unintended consequences of statin treatment guidelines or recommendations

through synthesis of high-quality observational and experimental data [20]. A simulation model was conceptualized and then developed in September 2018 to evaluate statin guidelines and recommendations because few available studies (1) were contemporary, (2) spanned ages specified by current guidelines or recommendations, and (3) precisely and validly measured ASCVD and T2D incidence within generalizable male and female multi-ethnic populations with long-term follow-up [21]. This study did not consider costs associated with statins, T2D, or ASCVD.

### Data sources and inputs

After conceptualizing the problem, the next step was to identify input data. In an attempt to maximize generalizability, we prioritized studies that included multi-ethnic (non-Hispanic African American and non-Hispanic white; capturing 73% of the U.S. population [22]) male and female statin-eligible adults aged 40–75 years, reflecting the ages specified by the AHA/ACC 2013 and 2018 guidelines [3, 4].

Demographic characteristics, 10-year ASCVD risk, and the number of adults eligible for statin therapy initiation were estimated using the biennial, cross-sectional, and nationally representative National Health and Nutrition Examination Survey (NHANES; waves 2007–2014), pooling 4 waves to ensure sufficient precision [23] (see S1 Text). NHANES collects demographic, nutritional, and health status information on a nationally representative probability sample of the U.S. civilian population instituted by the National Center for Health Statistics. We defined the primary prevention population as 40- to 75-year-old males and females of self-reported non-Hispanic African American or non-Hispanic white race/ethnicity who did not report a doctor or health professional diagnosis of ASCVD, T2D, or type 1 diabetes, who reported never taking cholesterol medications (assessed by self-report and medication inventory), and who had measured fasting LDL-C levels ≤ 190 mg/dL. Race/ethnicity-specific, sex-specific, and antihypertensive-therapy–specific predicted 10-year ASCVD risk for each participant of the primary prevention population was then calculated using the Pooled Cohort Equation (see S1 Text and S1 Table) [24]; note that we restricted our evaluation to statin guidelines or recommendations that used the Pooled Cohort Equation.

In the absence of nationally representative data measuring ASCVD and T2D incidence using validated protocols capturing undiagnosed disease and mortality [25], we leveraged data from the ongoing Reasons for Geographic and Racial Differences in Stroke (REGARDS) study (data used: 2003–2015). The REGARDS study is a contemporary, population-based, longitudinal cohort study designed to evaluate factors underlying the excess stroke mortality in the southeast versus other regions of the U.S. and among African American versus white adults (see S1 Text) [26]. Incident coronary heart disease (CHD) was defined based on medical records, signs and symptoms, diagnostic cardiac enzymes, or electrocardiographic changes consistent with myocardial infarction or a CHD death (S2 Table). Incident stroke was centrally adjudicated by physicians using the World Health Organization definition or by review of final reports from all available neuroimaging studies that were consistent with acute ischemia [27, 28]. Among participants without T2D at baseline who returned to visit 2, T2D incidence was defined by fasting glucose ≥126 mg/dl, non-fasting glucose >200 mg/dl, or use of glucose-lowering medication (S2 Table). Because the REGARDS study only included participants aged ≥45 years and given the comparability of estimated ASCVD and T2D incidence rates between ages 45 and 60, we assigned ASCVD and T2D incidence rates estimated in the 45- to 50-year age group to the 40- to 44-year age group.

Non-ASCVD mortality rates were obtained from the National Center for Health Statistics [29], which compiled death certificates filed in all 50 states and the District of Columbia.

Annual non-ASCVD deaths were defined excluding heart disease (International Statistical Classification of Disease, 10th edition codes [ICD-10]: I00–I09, I11, I13, and I20–I51) and cerebrovascular (ICD-10: I60–I69) deaths.

Primary prevention statin-associated ASCVD RRs were obtained from the Cholesterol Treatment Trialists' meta-analysis of 22 trials (statin treatment versus control), from which we abstracted separate published sex-specific RRs for males (RR = 0.64) and females (RR = 0.84) [30]. In the absence of consistent evidence of association between statins and non-ASCVD mortality [31], the statin–non-ASCVD mortality RR was set to 1.0.

Unlike the statin-ASCVD RRs, for which we only used published estimates from randomized controlled trials (RCTs), we considered meta-analytic estimates from RCTs and observational studies when quantifying statin-associated T2D risk: RR = 1.11 (RCTs only), RR = 1.32 (RCT and observational study pooled), and RR = 1.55 (observational studies only). This decision was motivated by previous reports documenting the potential for industry-funded RCTs to underreport adverse drug reactions [15, 32], the observation that confounding by factors that affect treatment assignment (i.e., "confounding by indication") is less likely in studies of adverse drug reactions [33], and the average shorter duration (2.6 years) of RCTs compared to observational studies [15].

Assignment of statin treatment was made at study baseline using predicted 10-year ASCVD risk as estimated by the Pooled Cohort Equation [34]. We assigned statin treatment if the predicted 10-year ASCVD risk equaled or exceeded the following 3 statin treatment guidelines or recommendations: ≥10% and the presence of at least one ASCVD risk factor (referred to hereafter as ASCVD risk >10%) [18]; ≥7.5% [3]; and ≥5% [3]. A fourth scenario in which no participant received statin treatment served as our reference group. The ACC/AHA 2018 guidelines were not evaluated due to the unavailability of several ASCVD risk enhancers in the continuous NHANES waves used to inform statin treatment decisions (i.e., persistently elevated LDL-C, persistently elevated triglycerides, preeclampsia, lipoprotein(a) (Lp[a]), ankle-brachial index, and coronary artery calcium were not measured) [4].

## Model overview

We contrasted statin-associated ASCVD and T2D incidence across 3 statin treatment guidelines or recommendations using state-transition Markov simulation models [35], which projected annual estimates of T2D and ASCVD incidence based on the previously defined parameters. For each annual cycle, statin-eligible populations could either remain alive and non-diseased (i.e., without ASCVD or T2D) or transition to having T2D, an ASCVD event (fatal or non-fatal), or a non-ASCVD death (S1 Fig). We did not simulate other disease states in addition to non-ASCVD death under our assumption that only ASCVD and T2D incidence were associated with statin use. As we were comparing incidence of T2D and ASCVD, once cohort members had T2D, an ASCVD event or non-ASCVD death, cohort members were removed from the population and were no longer simulated (see example in S1 Fig) [35]. The 1-year cycle was repeated 10 times for each age- (1-year) and sex-specific group and for each intervention scenario to project statin-associated 10-year ASCVD and T2D incidence. As our results project only the incident events for the primary prevention population over 10 years, we cannot make a statement about the population level ASCVD or T2D incidence in 10 years' time, as this estimate may depend on other factors outside of the scope of our model, e.g., migration patterns or progress in medical technology.

We assumed full implementation of statin treatment recommendations and 100% statin uptake at year 0, after which we assumed a 20% absolute annual decrease through year 4 (S2 Fig) [36]. At year 4, we assumed 20% of adults would continue taking statins through year 10.

Given the wide variation in assumed statin adherence found in the literature, we examined statin adherence further in sensitivity analyses.

Using these scenarios, we projected the absolute benefit of statin treatment as the number needed to treat (NNT) to prevent one ASCVD event (1/[statin-associated ASCVD risk difference]), with smaller values indicating a larger absolute risk reduction. The absolute harm of statin treatment was calculated as the number needed to harm (NNH[1/statin-associated T2D risk difference]) to cause one excess incident case of T2D, with larger values indicating that more people needed to receive statin treatment to cause one excess incident case of T2D [37]. In addition, we estimated the likelihood to be helped or harmed (LHH), defined as the ratio of the NNH to NNT; LLH values < 1 indicated that the number of incident cases of T2D incurred exceeded the number of ASCVD events prevented [38]. Statistical analyses were performed using STATA (College Station, TX), TreeAge Healthcare Pro Suite 2018 (TreeAge Software, Williamstown, MA), and SAS (Cary, NC) [39].

### Sensitivity analyses

To evaluate possible sources of heterogeneity, we projected the expected benefits and harms of statin treatment by sex and age [18, 40]. To examine the robustness of projections to different adherence patterns, we also projected the expected number of ASCVD events prevented and expected number of excess incident case of T2D incurred under 3 annual relative decrease in statin use of 25% and 50%, as well as full adherence (i.e., 0% decrease) across 10 years [36]. Finally, we examined the robustness of projections to variation in 2 sets of study input parameters (sex-specific statin-ASCVD RRs as well as sex- and age-specific ASCVD and T2D incidence rates) by performing 3 probabilistic sensitivity analyses (PSAs). Our first 2 PSAs considered uncertainty from each input parameter sets separately, and the third PSA considered uncertainty from each set of input parameter simultaneously. For each PSA, we ran 1,000 replications that sampled from the probability distribution of each parameter, which were determined based on 95% confidence intervals (S2 Table). For ASCVD and T2D incidence rates, we assumed a normal distribution for the mean as the estimate. For the statin-ASCVD RR, we assumed a lognormal distribution for the RRs [35]. Because the primary analyses already evaluated the influence of variation in statin-T2D RRs by considering 3 separate RRs, we did not consider this parameter in our PSA.

## Results

When weighted to the 2014 non-institutionalized, civilian U.S. population, our primary prevention population consisted of 61,125,042 adults (Table 1). Among the primary prevention population, a majority was female (58.5%) and non-Hispanic white (89.4%), with males and older adults having higher estimated 10-year ASCVD risks compared to females and younger adults (S4 Table). The proportion of the primary prevention population eligible for statin treatment initiation ranged from 21.8% (≥10% ASCVD risk threshold) to 45.6% (≥5.0% ASCVD risk threshold).

As the proportion of adults eligible for statin therapy increased, so did the number of ASCVD events prevented (Fig 1). Over 10 years, the ≥10% ASCVD risk threshold guideline was projected to prevent the fewest ASCVD events ($N$ = 103,009) (Fig 1, panel A), whereas the ≥5.0% ASCVD risk threshold was projected to prevent the largest number of ASCVD events ($N$ = 169,370).

When assuming a statin-associated T2D risk of 1.11, 10-year NNH projections were consistent across guidelines or recommendations, ranging from 444 to 446 (S3 Fig). These projections suggested that for all statin treatment guidelines or recommendations, the number of

**Table 1. Comparison of demographic and cardiovascular risk profiles for U.S. white and African American primary prevention populations aged 40–75 years overall and according to 3 statin treatment guidelines or recommendations.**

| Characteristic[a] | Primary prevention population[b] | 10-year ASCVD risk | | |
| --- | --- | --- | --- | --- |
| | | ≥10%[c] | ≥7.5% | ≥5.0% |
| *N* (% of total population) | 61,125,042 | 13,325,617 (21.8) | 18,613,696 (30.5) | 27,850,426 (45.6) |
| Female (%) | 35,758,150 (58.5) | 4,663,966 (35.0) | 6,924,295 (37.2) | 115,30,076 (41.4) |
| Non-Hispanic white (%) | 54,645,788 (89.4) | 11,486,682 (86.2) | 16,026,392 (86.1) | 24,341,272 (87.4) |
| Mean age (SD) | 54.7 (9.6) | 65.4 (6.9) | 63.6 (7.3) | 61.4 (7.8) |
| Mean high-density lipoprotein mg/dL (SD) | 55 (14.2) | 51 (15.3) | 52 (15.3) | 51 (15.1) |
| Mean total cholesterol mg/dL (SD) | 204 (39.4) | 204 (39.1) | 205 (38.9) | 207 (38.6) |
| Mean systolic blood pressure mmHg (SD) | 122 (19.6) | 134 (18.8) | 133 (18.4) | 129 (18.0) |
| On hypertension treatment (%) | 209,04764 (34.2) | 7,968,719 (59.8) | 10,405,056 (55.9) | 13,953,063 (50.1) |
| Current smokers (%) | 12,591,759 (20.6) | 3,877,755 (29.1) | 5,323,517 (28.6) | 7,714,568 (27.7) |

[a]Weighted means and proportions.

[b]After excluding prevalent statin users and adults with self-reported ASCVD and T2D and upweighting to the 2014 U.S. population.

[c]In the presence of at least one ASCVD risk factor.

**Abbreviations:** ASCVD, atherosclerotic cardiovascular disease; T2D, type 2 diabetes

ASCVD events prevented was at least twice as large as the number of incident cases of T2D incurred (LHH range 2.26–2.90; NNT range 155–215) (Fig 1, panels A and D; S3 Fig).

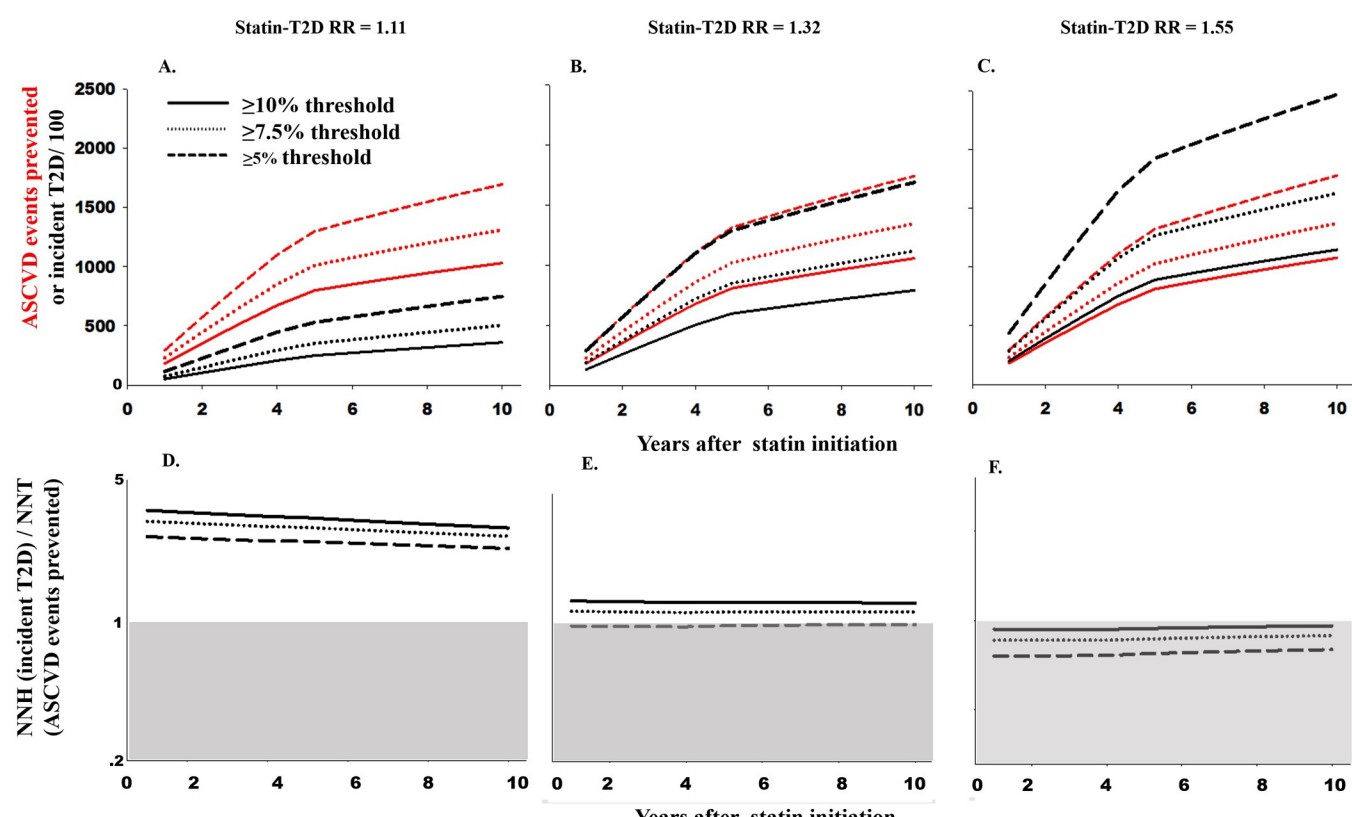

**Fig 1.** Cumulative number of events of ASCVD and T2D (panels A–C) and LHH (NNH/NNT; panels D–F) associated with 3 statin treatment guidelines or recommendations among a primary prevention population of 61,125,042 eligible U.S. African American and white adults in 2014. The shaded area in panels D–F conveys when the NNH > NNT. ASCVD, atherosclerotic cardiovascular disease; LHH, likelihood to be helped or harmed; NNH, number needed to harm; NNT, number needed to treat; RR, relative risk; T2D, type 2 diabetes.

However, projections of absolute and relative harm were sensitive to the assumed statin-associated T2D RR. When the statin-associated T2D RR was increased to 1.32, NNHs decreased to 198–202 and the relative benefits of statin treatment decreased (LHH range: 1.03–1.30; NNT range 155–209) (Fig 1, panels B and E; S3 Fig). Sensitivity analyses that varied adherence to statin treatment resulted in proportional decreases in the number of ASCVD events prevented and incident cases of T2D incurred (S6 Fig), although results for LHH projections remained consistent.

We next examined the benefits and harms of statin treatment by sex. Across all scenarios, females received lower absolute benefits and incurred a higher relative burden of adverse events compared to males (Figs 2 and S4). The absolute and relative benefits of statin treatment guidelines and recommendations also were more variable in females compared to males. For example, when assuming a statin-T2D RR = 1.11, one ASCVD event was prevented for every 196–254 females treated (LHH range: 1.74–2.40; NNH range 430–478) across the 3 statin treatment guidelines or recommendations. For males, one ASCVD event was prevented for every 110–131 males treated (LHH range: 2.55–3.00; NNH range 331–334) across the 3 statin treatment guidelines or recommendations. When assuming a statin-T2D RR = 1.32, LHH ranged from 0.77 to 1.1 in females but remained above 1 (LHH range: 1.18–1.43) for males. Consistent with our findings, PSA indicated that estimates in females were more uncertain than estimates in males or the total population (S7 Fig).

We also examined benefits and harms of statin treatment by age. Regardless of scenario, the absolute and relative benefits of statin treatment were lowest in populations aged 40–50 years and highest in populations aged 71–75 years (Figs 3 and S8). As an example, when assuming a statin-associated T2D RR of 1.11, adults aged 40–50 received the lowest absolute benefits of statin

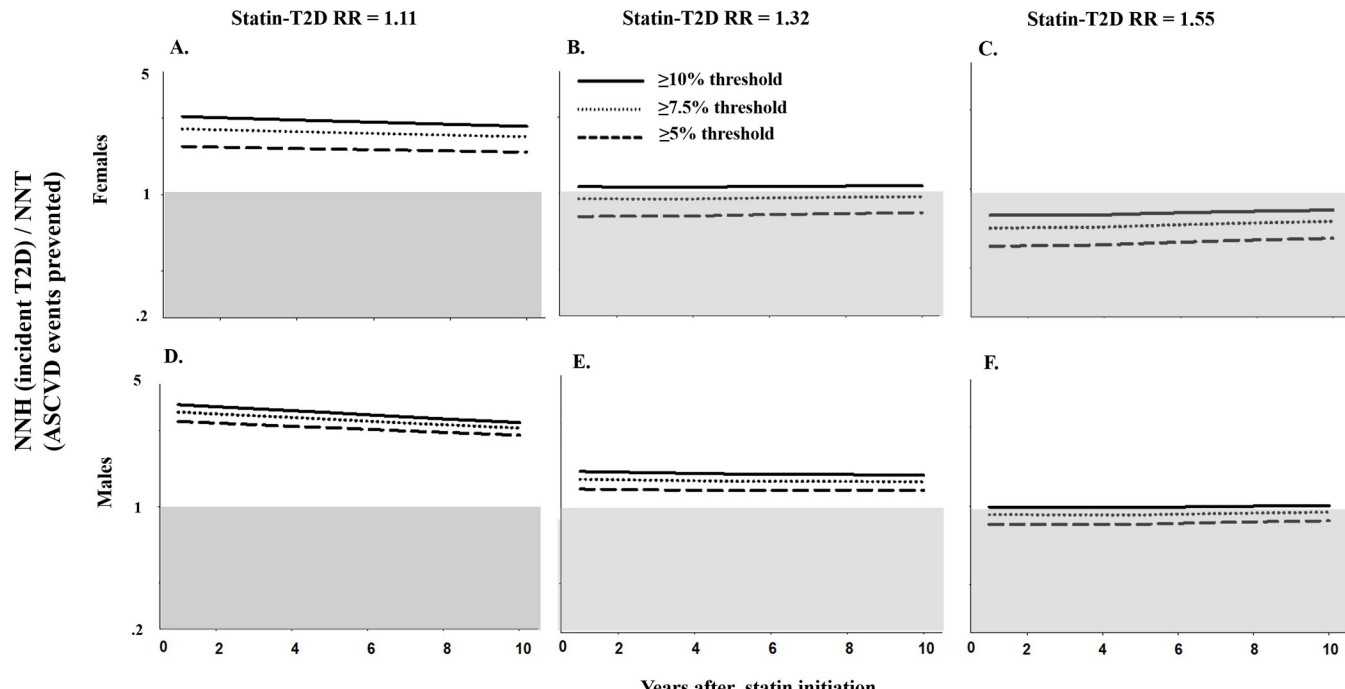

**Fig 2.** LHH (NNH/NNT) among females (panels A–C) and males (panels D–F) associated with 3 statin treatment guidelines or recommendations among a primary prevention population of 61,125,042 eligible U.S. African American and white adults in 2014. Shaded area describes when NNH > NNT. ASCVD, atherosclerotic cardiovascular disease; LHH, likelihood to be helped or harmed; NNH, number needed to harm; NNT, number needed to treat; RR, relative risk.

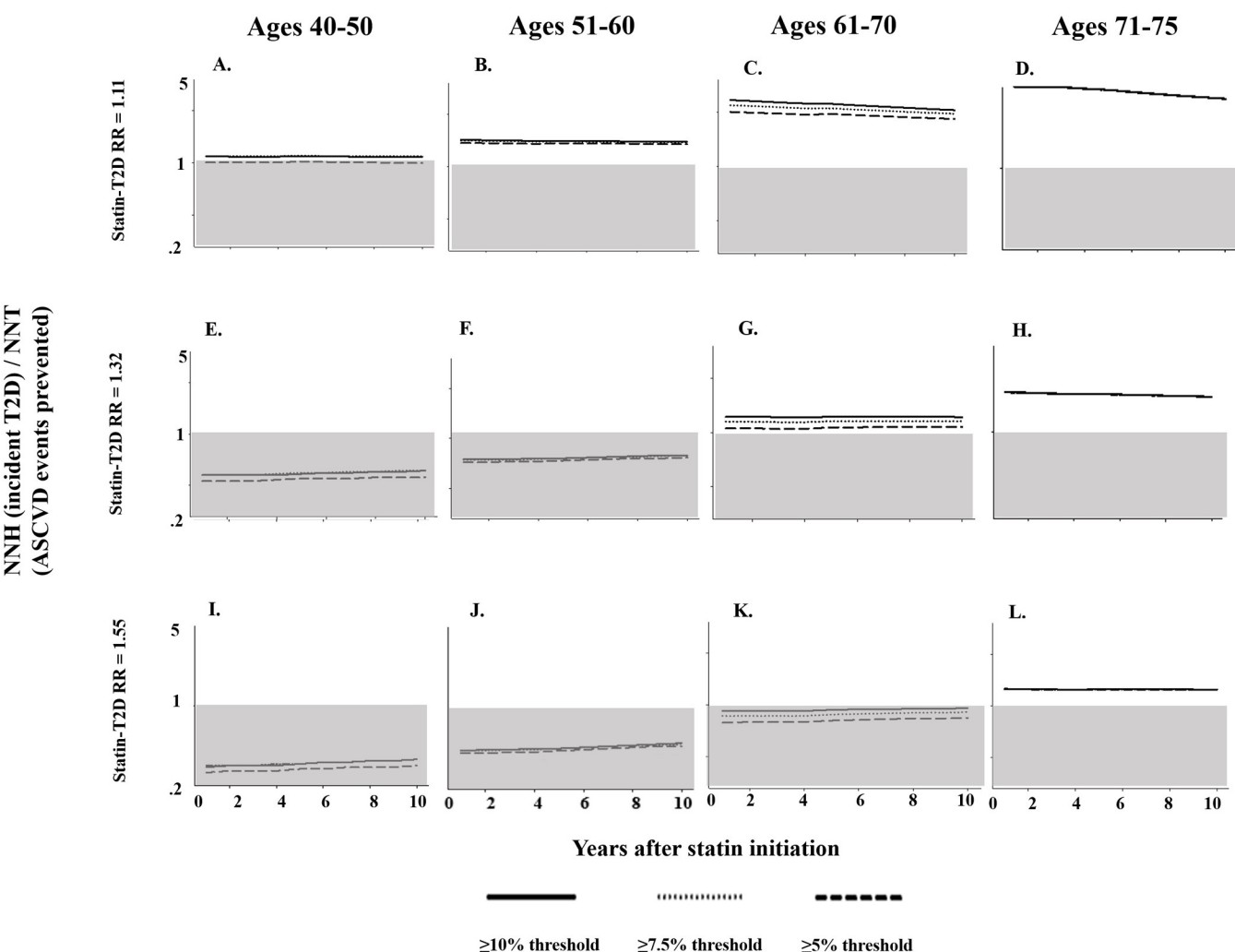

**Fig 3.** LHH (NNH/NNT) associated with 3 statin treatment guidelines or recommendations among 40–50 (panels A, E, I), 51–60 (panels B, F, J) 61–70 (panels C, G, K), and 71–75 (panels D, H, L) baseline age groups among a primary prevention population of 61,125,042 eligible African American and white U.S. adults in 2014. ASCVD, atherosclerotic cardiovascular disease; LHH, likelihood to be helped or harmed; NNH, number needed to harm; NNT, number needed to treat; RR, relative risk; T2D, type 2 diabetes.

treatment (NNT range: 322–378) and incurred the highest relative burden of adverse events (LHH range: 1.0–1.14) compared to other age groups (Fig 3, panel A). When a statin-associated T2D RR of 1.32 was assumed, adults aged 40–50 continued to receive the lowest absolute benefits of statin treatment (NNT range: 313–367), with the relative burden of adverse events suggesting that every ASCVD event prevented was associated with at least 2 incident cases of T2D (LHH range: 0.43–0.49) (Fig 3, panel E). In contrast, adults aged 71–75 years received the highest absolute benefit (NNT range: 106–107) and the lowest relative burden of adverse events (LHH range: 1.92–1.93), projections that showed very modest differences across statin treatment guidelines or recommendations or assumed statin-associated T2D RR (Fig 3; panel D).

## Discussion

In this study, we examined the net effects of statins across 3 treatment guidelines or recommendations in a contemporary, biracial adult primary prevention population. We projected

that 13 to 28 million non-Hispanic African American and white adults would be newly eligible for statin treatment, among whom one ASCVD event would be prevented for every 155–197 adults treated. Benefits of statin treatment were even more pronounced in males and older populations compared to females and younger populations. Quantifying harms associated with statin treatment was more complex, as changes in statin-associated T2D risk produced large differences in projected harms. Overall, these results suggest that further efforts are needed to better quantify statin-associated T2D risk across a range of populations, particularly female and younger adult populations for whom statin treatment may introduce a large relative burden of adverse events.

One major challenge for research examining the net effects of statins is comparing intended benefits and unintended harms, which may not be equivalent. Here, the broad clinical spectrum associated with ASCVD that spans a biomarker-diagnosed silent myocardial infarction to a disabling stroke to sudden death would have different implications for patients as well as different clinical courses compared to a new diagnosis of T2D. That being said, as the goal of statin therapy is to prevent ASCVD, and considering that T2D is a clinically important harm associated with statin therapy that has a similar increase in risk as a coronary event for a subsequent ASCVD event [41], this contrast remains a relevant comparison. Particularly of interest are scenarios assuming higher statin-associated T2D RR that compared females versus males and younger versus older populations. In these scenarios, for females and younger populations, the number of incident cases of T2D incurred were often greater, sometimes by several orders of magnitude, then the number of ASCVD events prevented. One way forward may be to incorporate ASCVD and T2D risk prediction models in patient-tailored decision-support tools to select treatment options that balance risks and benefits of statins. Yet, despite the fact that all guidelines emphasize shared decision making, none currently provide the necessary tools [42].

Our projections also suggested that males and older adults received greater benefits from statin treatments compared to females and younger adults. Understanding the risk–benefit profile of statin treatment in younger and female populations is important, given the large proportions of females eligible for statin treatment (35%–41%) as well as uncertainties surrounding adverse effects associated with long-term statin use [3]. Differences between males and females in the net benefits of statin treatment may reflect several factors, including estimates of statin-associated ASCVD risk, which showed a more protective effect in males than females. Heterogeneity by sex in age-specific ASCVD incidence rates also is long-described [43], which could further decrease the net benefits of statin therapy for ASCVD reduction. Interestingly, available studies also support the potential for net harms to be greater in females than males [15, 16, 44, 45]. For example, the Justification for the Use of statins in Prevention: an Intervention Trial Evaluating Rosuvastatin (JUPITER) trial reported a 50% increase in physician-reported T2D in females compared to males, corresponding to an estimated 11 incident T2D diagnoses per 1,000 females taking statins over 1.9 years [16]. However, our simulation assumed a constant statin-associated T2D RR by sex. Additional efforts that enable quantification of the relative and absolute net effects of statin treatment by sex are warranted, particularly in light of disparities in statin treatment by sex [(46], the continued under-representation of female participants in RCTs supported by the US Food and Drug Administration for approval of new molecular entities [46, 47], and the limited number of studies that have examined evidence of heterogeneity in statin-associated T2D risk by sex [30, 48].

In addition, the youngest age groups, which composed 26% of the primary prevention population, did not realize the same statin-associated benefits as the oldest age groups. The importance of research quantifying the net benefits of statins across adulthood is underscored by the fact that statin initiation in younger ages may be associated with decades of statin treatment, despite mixed evidence of long-term effectiveness or greater risk reduction in younger

populations [30]. For example, the Anglo-Scandinavian Cardiac Outcomes Trial-Lipid Lowering Arm (ASCOT-LLA) found a statistically significant reduction in ASCVD associated with statins approximately 2 years post follow-up (hazard ratio [HR] = 0.64 [95% CI: 0.53–0.78]) but not 11 years post follow up (HR = 0.89 [95% CI: 0.72–1.11]) [49, 50]. Among additional trials, information on post-trial statin use among those initially randomized to statins or placebo was not always known, further suggesting that the association between long-term, persistent use of statins and adverse events remains incompletely quantified [49, 51].

Our results also underscore the influence of assumed statin-associated T2D risk when quantifying statin net benefit. Although the majority of published studies examining statin-associated harms and benefits leveraged statin-associated T2D RRs from RCTs [17], RCTs may underestimate adverse drug effects due to under-reporting of harms or limited follow-up time [52, 53]. A modest—albeit growing—body of literature also has examined the implications of poor external validity in RCTs, which has the potential to limit transportability of RCT-derived RR estimates to external populations [53]. Specifically, if factors including age, sex, or health characteristics modified the association between statin treatment and T2D, then application of such estimates to populations with different age, sex, or health characteristic has the potential to inaccurately quantify statin-associated T2D (or ASCVD) risk [54]. In this study, RCTs used to quantify statin-associated T2D risk enrolled older (mean age = 63.6 years) and predominantly male (64.1% male) populations compared to the statin-eligible primary prevention population simulated herein (41.5% male, mean age = 54.7 years) or observational studies included in prior meta-analyses (48.5% male, mean age = 57.4 years). Although it remains difficult to anticipate the magnitude by which differences in population characteristics affect estimates of statin-associated T2D risk, prior studies reporting potential heterogeneity by age, sex, and health characteristics support evaluating a range of potential RR estimates rather than relying on a single estimate [10, 15, 55].

Despite many strengths, there are limitations that merit consideration. First, we were unable to examine other adverse events associated with statin use, including rhabdomyolysis [56–58], although the rarity of the event (potentially impacting 546–1,344 adults who were newly eligible for treatment in our study [56]) likely resulted in a very modest underestimate of harm. We also did not consider additional potential benefits of statin treatment, including studies reporting a decreased risk of breast cancer recurrence among females treated with statins [59]. This decision reflected our prioritization of ASCVD, for which the evidence base was the strongest, although future simulations may consider breast cancer recurrence or other potential benefits or harms as evidence accumulates [59–61]. Second, we limited our study to projecting incidence in a primary prevention populations 40–75 years of age of non-Hispanic African Americans or white race/ethnicity given limited input data (e.g., T2D and ASCVD incidence) in other racial/ethnic groups. Although this decision may limit generalizability, 73% of the U.S. population are non-Hispanic African American or white [22]. We also reported projections in the total population and not by race/ethnicity, anticipating imprecise model inputs by race/ethnicity. Future work may wish to expand simulations to include additional populations and evaluate heterogeneity by race, ethnicity, or other potentially modifying factors. Third, our estimate of T2D incidence is based on REGARDS participants returning for the second visit and may be affected by participant attrition. We estimated T2D incidence using REGARDS data because other available sources, e.g., national incidence estimates [62], were based on self-report, which is moderately sensitive and cannot capture the large burden of undiagnosed diabetes in the U.S. [63, 64]. Fourth, 2 model inputs included non-U.S. data: meta-analyzed estimates of statin-T2D RRs and meta-analyzed estimates of the statin-ASCVD RRs. For the former estimate, our previous meta-analysis did not detect significant heterogeneity by country of residence ($P > 0.05$), supporting pooling. For the latter estimate, we could not identify a meta-analysis that tested for

heterogeneity by country of residence, although prior population-based studies conducted before widespread statin use suggested comparable LDL-C levels in European and North American adults. However, the degree to which heterogeneity from variation in the epidemiology of ASCVD by country of residence affects estimates of statin-associated ASCVD RRs cannot be evaluated in depth [65, 66]. Finally, our prioritization of guidelines based on the Pooled Cohort Equation led us to exclude the National Institute for Health and Care Excellence (NICE) [67], the Canadian Cardiovascular Society (CCS) [68], and the European Society of Cardiology (ESC)/ European Atherosclerosis Society (EAS) [69] guidelines. However, recent reports have suggested overlap between guidelines, and comparable estimated NNTs have been reported for the CCS, ACC/AHA, and NICE guidelines as well as the U.S. Preventive Services Task Force (USPSTF) and ESC/EAS guidelines [70]. We also did not explicitly consider the ACC/AHA 2018 guideline given data limitations. However, both the 2013 [3] and 2018 [4] ACC/AHA guidelines use the Pooled Cohort Equation with 5% and 7.5% treatment thresholds as starting points for considering statin therapy among adults age 40–75 being evaluated for primary ASCVD prevention. The USPSTF suggests a threshold of 10% for initiating statin therapy for primary prevention [18]. Thus, the 5%, 7.5%, and 10% thresholds for statin eligibility aligns appropriately with current guidelines and physician practice [71] when determining cut-points for initiating statin therapy.

In conclusion, this simulation study adds to a growing body of literature examining the net effects of statins in primary prevention populations. Our results suggest that the highest relative burden of T2D occurred among female and younger adult populations and highlight areas in which additional clinical and public health research is needed.

## Supporting information

**S1 Table. Estimation of race- and sex-specific ASCVD risk using the ASCVD Pooled Cohort risk equations.** ASCVD, atherosclerotic cardiovascular disease.
(DOCX)

**S2 Table. Model input parameters stratified by 5-year age groups and sex.**
(DOCX)

**S3 Table. Markov model parameters.**
(DOCX)

**S4 Table. U.S. African American and white primary prevention populations aged 40–75 years according to 10-year ASCVD thresholds stratified by 5-year age groups and sex.** ASCVD, atherosclerotic cardiovascular disease.
(DOCX)

**S1 Fig. Markov model conceptual diagram for projections of ASCVD, T2D, and non-ASCVD mortality among an eligible primary prevention population.** Rectangles correspond to disease states, and arrows represent the allowed transitions. Absorbing states are shaded. ASCVD, atherosclerotic cardiovascular disease; T2D, type 2 diabetes.
(TIF)

**S2 Fig. Proportion of adults adhering to statin treatment guidelines or recommendations over 10 years among a projected population of 61,125,042 eligible U.S. African American and white adults in 2014.**
(TIF)

**S3 Fig. NNT or NNH associated with 3 statin treatment guidelines or recommendations among a projected population of 61,125,042 eligible U.S. African American and white**

**adults in 2014.** NNH, number needed to harm; NNT, number needed to treat.
(TIF)

**S4 Fig. NNT or NNH among females (panels A–C), and males (panels D–F) associated with 3 statin treatment guidelines or recommendations among a projected population of 61,125,042 eligible U.S. African American and white adults in 2014**. NNH, number needed to harm; NNT, number needed to treat.
(TIF)

**S5 Fig. Cumulative number of events of ASCVD and T2D among females (panels A–C) and males (panels D–F) associated with 3 statin treatment guidelines or recommendations from a projected population of 61,125,042 eligible U.S. African American and white adults in 2014**. ASCVD, atherosclerotic cardiovascular disease; T2D, type 2 diabetes.
(TIF)

**S6 Fig. LHH (NNH/NNT) associated with 3 statin treatment guidelines or recommendations among a projected population of 61,125,042 eligible U.S. African American and white adults in 2014.** Grey line describes threshold when NNH > NNT. Statin-T2D RR = 1.11. LLH, likelihood to be helped or harmed; NNH, number needed to harm; NNT, number needed to treat; RR, relative risk; T2D, type 2 diabetes.
(TIF)

**S7 Fig. Boxplots of the uncertainty intervals for the LHH (NNH/NNT) associated with the 7.5% ASCVD 10-year risk threshold statin treatment guideline among a projected population of 61,125,042 eligible U.S. African American and white adults in 2014.** Uncertainty was quantified through 3 PSAs: 2 PSAs that considered uncertainty from each of the input parameters separately and a third PSA that considered uncertainty from each input parameter simultaneously. Gray shading indicates the portions of the uncertainty intervals for which the NNH exceeds the NNT. ASCVD, atherosclerotic cardiovascular disease; LLH, likelihood to be helped or harmed; PSA, probabilistic sensitivity analysis; NNH, number needed to harm; NNT, number needed to treat.
(TIF)

**S8 Fig. NNT or NNH among 40- to 50-year-olds (panels A–C), 51- to 60-year-olds (panels D–F), 61- to 70-year-olds (panels G–I), and 71- to 75-year-olds (panels J–L) associated with 3 statin treatment guidelines or recommendations among a projected population of 61,125,042 eligible U.S. African American and white adults in 2014**. NNH, number needed to harm; NNT, number needed to treat.
(TIF)

**S1 STROBE checklist. STROBE Statement—Checklist of items that should be included in reports of cohort studies.** STROBE, Strengthening the Reporting of Observational Studies in Epidemiology.
(DOCX)

**S1 Text.**
(DOCX)

## Acknowledgments

The authors thank the other investigators, the staff, and the participants of the REGARDS Study for the valuable contributions. A complete list of participating REGARDS investigators and institutions can be found at https://www.uab.edu/soph/regardsstudy/.

## Author Contributions

**Conceptualization:** Joseph C. Engeda, Stefan K. Lhachimi, Wayne D. Rosamond, Jennifer L. Lund, Thomas C. Keyserling, Lisandro D. Colantonio, Christy L. Avery.

**Formal analysis:** Joseph C. Engeda, Stefan K. Lhachimi, Thomas C. Keyserling.

**Investigation:** Joseph C. Engeda, Thomas C. Keyserling, Christy L. Avery.

**Methodology:** Stefan K. Lhachimi, Jennifer L. Lund.

**Software:** Stefan K. Lhachimi.

**Supervision:** Christy L. Avery.

**Writing – original draft:** Joseph C. Engeda, Christy L. Avery.

**Writing – review & editing:** Wayne D. Rosamond, Jennifer L. Lund, Monika M. Safford, Lisandro D. Colantonio, Paul Muntner.

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
