## [Decision Letter · Decision Letter 0]

28 Apr 2020

Dear Dr. Engeda,

Thank you very much for submitting your manuscript "Projections of incident atherosclerotic cardiovascular disease and incident type 2 diabetes across evolving statin treatment guidelines and recommendations" (PMEDICINE-D-19-04150) for consideration at PLOS Medicine. 

[LINK]

In light of these reviews, I am afraid that we will not be able to accept the manuscript for publication in the journal in its current form, but we would like to consider a revised version that addresses the reviewers' and editors' comments. Obviously we cannot make any decision about publication until we have seen the revised manuscript and your response, and we plan to seek re-review by one or more of the reviewers. 

We expect to receive your revised manuscript by May 19 2020 11:59PM. Please email us (plosmedicine@plos.org) if you have any questions or concerns.

We look forward to receiving your revised manuscript. 

Sincerely,

Emma Veitch, PhD

PLOS Medicine

On behalf of Clare Stone, PhD, Acting Chief Editor,

PLOS Medicine

plosmedicine.org

*Please revise your title according to PLOS Medicine's style. Your title must be nondeclarative and not a question, beginning with the main concept. The study design (eg in this case - "modelling study," etc.) should appear in the subtitle (ie, after a colon).

*Please structure your abstract using the PLOS Medicine headings (Background, Methods and Findings, Conclusions) -"methods and findings" should be a single subsection. Please also rephrase some of the sections so these are written as full sentences rather than sentence fragments ("to compare.." etc).

*In the last sentence of the Abstract Methods and Findings section, please describe the main limitation(s) of the study's methodology.

*At this stage, we ask that you include a short, non-technical Author Summary of your research to make findings accessible to a wide audience that includes both scientists and non-scientists. The Author Summary should immediately follow the Abstract in your revised manuscript. This text is subject to editorial change and should be distinct from the scientific abstract. Please see our author guidelines for more information: https://journals.plos.org/plosmedicine/s/revising-your-manuscript#loc-author-summary

*Ideally please change the referencing format (this should be simple if referencing software has been used) to change citation callouts in the text to numbers in square brackets (ie, [1, 2] etc). Many thanks

*Did your study have a prospective protocol or analysis plan? Please state this (either way) early in the Methods section.

Comments from the reviewers:

Reviewer #1: The aim of this study by Engeda and colleagues was to investigate (and project) the potential impact of statin use in primary prevention, that is, reducing the risk of atherosclerotic cardiovascular disease (ASCVD) on the one hand, and the potential harm - risk of incident type 2 diabetes mellitus in a population of adult Americans aged 40 - 75 years. A Markov model was developed and used to compare the projected the impact (benefit and harms) on population groups stratified by 10-year ASCVD risk, based on American College of Cardiology (ACC)/ American Heart Association (AHA) guidelines. 

Authors report that in the overall cohort, the number of averted ASCVD events outweigh the number incident type 2 diabetes mellitus cases from statin treatment for primary prevention, with a likelihood to help or harm of 2.26 - 2.90. The magnitude of this impact was greater in men and older people as opposed to women and younger people with more modest impact. Their findings are sensitive to the varied relative risks of statin use and risk of type 2 diabetes mellitus. Their findings are interesting as it provides insights to a less-well investigated relationship explored in the same population. 

I have a few comments/questions on the methods and reporting of results that require clarification.

Methods

1) Page 6, "The first step in constructing the simulation model was to collect input data." I understand authors are referring to model development when they talk about 'constructing' a simulation model? If that be the case, it appears this does not quite sit well with the International Society for Pharmacoeconomics and Outcomes research (ISPOR) guidelines for good research practices (See: Med Decis Making. 2012 Sep-Oct;32(5):678-89.), which suggest that conceptualization and model development to answer the decision question should precede the search of input data. The model structure should therefore not [initially] be dictated by the range of available data but rather the other way around. Admittedly, this goes through an iterative process before finally adopting the final model. Authors might want to revise this statement to better reflect their process aligns with guidelines? 

2) Page 8, "Primary prevention statin-associated ASCVD RRs were obtained from the Cholesterol Treatment Trialists' meta-analysis of 22 trials (statin treatment versus control), from which we abstracted separate RR point estimates for males (RR=0.78) and females (RR=0.84).(28)" I have some questions in relation to this;

a) A good number of the trials included in this meta-analysis were conducted in the UK and Europe. Authors may want to comment on the transferability/applicability of these estimates to the U.S., given there's likely differences in the epidemiology of, and treatment of CVD events between these countries and the U.S.? Was there not pooled evidence from the literature with U.S specific data that could be used instead?

b) Authors purport to have extracted point estimates of relative risks for males and females from this study. Authors should clarify how this was done. In addition, it is unclear if there was any uncertainty e.g. 95% confidence intervals for these estimates. I would expect some uncertainty around the relative risk of events, which needs to be accounted for in the probabilistic sensitivity analysis, to get a comprehensive sense of the risk/impact. Could authors clarify how this was managed?

c) Relative risk of vascular events would vary by age, e.g. differ for 40-50yrs, 51-60yrs, 61-70yrs, etc. Using just a single point estimate of RR is likely to undermine this heterogeneity [at least by age]. In a later part of the results, authors discuss variation by age, but it is unclear or not immediately apparent how this was approached, at least, in terms of the relative risks. Could authors comment on this with respect to their modelling (possible limitation to be discussed)? 

3) Page 9, Model overview: "For each annual cycle, statin-eligible populations could either remain alive and non-diseased or transition to having T2D, an ASCVD event, or a non-ASCVD death". Did the health state 'ASCVD event' include fatal and non-fatal cases? Otherwise, how were the fatal ASCVD events modelled?

4) Regarding heterogeneity, it would be interesting to see if there was a differential in the impact of statin treatment on ASCVD events between Black Americans, Hispanic and non-Hispanic whites. In addition, variation in impact by socio-economic status (SES). I wonder if authors considered exploring these (at least in sensitivity analysis) as there are likely to be differences in risk by ethnicity and SES.

5) The paper uses the REGARDS study for input data on T2D and ASCVD incidence. There are some issues with this: 

a) Authors should clarify why the data from REGARDS was chosen in preference to other data. For example, there seems to be data available from the Centre for Disease Control (page 19 and reference 58). 

b) While REGARDS is a national survey, it is unclear if it is nationally representative, given that the study aim was to include 30% of participants from the "Stroke Belt" states. Please confirm the study is nationally representative. 

c) The REGARDS study only included participants from ages 45 years, but the paper starts with populations at age 40 and uses the data from the 45 - 50 age group for the 40 - 45 group and "assessed different specifications via meta analyses". Please clarify what this means.

6) The paper uses 10-year ASCVD risk to determine eligibility for statin treatment. This reflects the 2013 Guidelines but does not reflect the 2018 AHA/ACC Guideline on the Management of Blood Cholesterol (2018 Guidelines), which use other parameters to determine statin eligibility. Further, it is unclear whether the use of 10-year ASCVD risk to determine eligibility for statin treatment reflects current practice. Authors should provide information to demonstrate that the statin eligibility criteria used in the intervention groups reflect current practice. This is important to demonstrate that the population eligible for statins in the intervention groups accurately reflect those people that would be eligible for statins in the population. 

7) The current analysis seems not to have performed an uncertainty analysis? I would assume authors would consider uncertainty distributions around key input parameters e.g. relative risk, to allow for a probabilistic sensitivity analysis?

Results and discussion

8) Page 16, paragraph 1: The paper states "one ASCVD event would be prevented for every 155 - 197 adults treated" and "benefits of statin treatment were even more pronounced in males and older populations compared to females and younger populations". These statements rely on the NNB produced by the model. However, the results section does not disclose the NNB - it focuses on the changes in ASCVD events prevented between different guidelines and the changes to LHH between different statin-associated T2D RRs, sex and age groups. The authors should include in the results the NNB, including how this varies by age and sex, in order to support this part of the discussion. 

9) Page 20, paragraph 1: The paper states "In addition, we did not consider the 2018 guidelines given data limitations although we anticipate projections of these guidelines would fall between 5% and 7.5% ASCVD risk thresholds we considered, providing some degree of information." This sentence appears incomplete. Authors should clarify what information is provided. For example, do these projections provide information about the risk-benefit analysis under the 2018 Guidelines? The authors should clarify the basis of their anticipated projections - how were they reached? The cited reference (62) does not seem to support the statement and is relevant to the previous point (comparison of other published guidelines to the 2013 Guidelines). 

Minor comments and considerations

* The calculation of statin eligibility for the intervention groups relies upon calculation of the 10-year ASCVD risk using the Pooled Cohort Equation. The Equation is used to estimate 10-year ASCVD risk for an individual. Authors explain in the supplementary material (page 3) how they calculate a 10-year ASCVD risk at population level. The authors should clarify that section to make clear how this is achieved and avoid confusion. 

* Page 6, end of first paragraph: The reference should be amended to include reference 3 and 4 because the sentence refers to both the 2013 and 2018 Guidelines. 

* Page 9, paragraph 2 & page 10, paragraph 2: Can authors provide some justification for their choice of adherence rates? including those in the sensitivity analyses. The cited study (reference 34) aimed to identify factors that predict adherence to statins rather than the adherence rates. However, it does not seem to include data about adherence rates (page 1413 of reference 34). Authors may want to provide clarification on how this reference informed their adherence rates.

* Page 16, paragraph 1: "overall, these results suggest that further efforts are needed to more precisely characterize populations for whom statin treatment may introduce a large burden of adverse effects". Authors should clarify this sentence to provide a more specific conclusion that follows from the preceding sentences. The problem alluded to in the preceding sentence is the lack of a precise estimate of statin-associated T2D risk. If that is correct, a more pertinent conclusion might be that further efforts are required to estimate this risk. 

* Page 16, paragraph 2: "particularly of interest are scenarios where the number of incidence cases of T2D incurred were greater than, sometimes by several orders of magnitude, the number of ASCVD events prevented". The author should clarify whether they are referring to scenarios produced by this study and, if so, which scenarios. 

* Page 19 - 20: In response to the limitation of not using multiple guidelines, authors state: "recent reports have suggested overlap between guidelines considered herein with other published guidelines. For example, comparable estimated NNTs have been reported" between other published guidelines and those used by the study. To demonstrate the suggested overlap, the authors may want to consider comparing these other published guidelines and the 2013 Guidelines by reference to the populations that would be eligible for statin therapy rather than the effects of statin therapy (i.e. NNTs). For example, to compare the 2018 Guidelines and the 2013 Guidelines, the authors refer to projections of the ASCVD risk thresholds produced by the 2018 Guidelines. Perhaps a similar projection could be made for the other published guidelines.

Supplementary material

1) Authors state, "The definition of stroke in the REGARDS study was: Prevalent stroke was defined as a positive response to either "Were you ever told by a physician that you had a stroke?" or "Were you ever told by a physician you had a mini-stroke or TIA, also known as a transient ischemic attack?" I anticipate that the risk of TIA and the risk of ischaemic stroke are different. Any considerations in this risk differential?

2) In S4 Table, frequencies seem to be in parenthesis and percentage out, contrary to what authors refer to as for example, Male (%). Please, consider adjusting.

References

Reference 15, 28 and 58 and 62 seem incomplete.

Reviewer #2: This is a well conducted study on the projections of incident ASCVD and incident type 2 diabetes using different assumptions of statin treatment recommendations. The study design, datasets, statistical methods and analyses, and presentation (tables and figures) and interpretation of the results are mostly adequate. However, there are still a few issues needing attention.

1) This is a simulation study using the Markov model, popular in the health economics field, which is fine. It's based on many assumptions and also the parameters estimated from the existing datasets. The key question is the believability and robustness of the model, which we don't really know. Are there any validation procedures for the proposed models? In other words, to what degree, should we trust the model that can reflect what's going on in the real world and in the future?

2) Although the Markov model can be used for simulation, it seems a bit simplistic. A few issues remains. Firstly, it's highly likely the model will rely on the projected prevalence of ASCVD and T2D in the next 10 years. Not sure how exactly the authors did it. It seems that they used the REGARDS study data to estimate? Then what prediction models were used for the projection, linear, Poisson or exponential? They are many ways to predict the prevalence using past and future information based on different models. Some good examples can be seen from the latest Global Burden of Disease (GBD) studies. Robust and reliable estimates of future prevalence of the events will have a big impact on the simulation results, and at least this should be discussed in the limitation. Secondly, in the Markov model, for different transition states, what happens if a person is still alive but developed diseases other than ASCVD or T2D later? It seems they were not assigned to any state in the study.

3) The presentation of Table 1 needs to improve and to use standard format. For continuous variables such as age and total cholesterol, they should be summarised as mean and SD if normal distribution, or median and IQR if non-normal distribution. For categorical data, they need to be summarised as count and percentage.

Reviewer #3: 

Projections of incident atherosclerotic cardiovascular disease and incident type 2 diabetes across evolving statin treatment guidelines and recommendations

In this study, Dr Engeda and co-workers have estimated the CVD risk reduction from statins used against induced Type 2 diabetes in adults free of prior CVD, eligible to statins use according to contemporary guidelines. Using simulation models (Markov models) they found that the number of incident CVD averted was at least twice as higher the number of incident diabetes induced. Women and younger adults had the lowest absolute benefits and the investigators concluded that these were areas in need for additional clinical and public health research.

While the question investigated has merit, the approach used (simulation), which is essentially based on assumptions, has limitations, many of which are highlighted by the investigators. Where the study really falls short in my view is that the investigators should have included possible scenario to mitigate diabetes risk from statin use. Just like they have used absolute risk model to predict CVD risk, similar models exist to predict incident diabetes risk. Furthermore, lifestyles and other interventions the reduced the unset of diabetes in people at risk are also know. The investigators could therefore consider accounting for the impact interventions to prevent diabetes in those at high risk at baseline, on the future risk of type 2 diabetes in people started on statins. The reality is that, in spite of the authors' findings, chances of withholding statins in eligible patients because of the fear of diabetes risk are very low, and therefore additional focus should be on how to mitigate the risk of diabetes in people started on statin. Other considerations: Incident diabetes was based on history, fasting or random glucose, which is not optimal.

[LINK]

---

## [Decision Letter · Decision Letter 1]

3 Jul 2020

Dear Dr. Engeda,

Thank you very much for re-submitting your manuscript "Projections of incident atherosclerotic cardiovascular disease and incident type 2 diabetes across evolving statin treatment guidelines and recommendations: A modelling study" (PMEDICINE-D-19-04150R1) for review by PLOS Medicine.

I have discussed the paper with my colleagues and the academic editor and it was also seen again by reviewers. I am pleased to say that provided the remaining editorial and production issues are dealt with we are planning to accept the paper for publication in the journal.

[LINK]

We look forward to receiving the revised manuscript by Jul 10 2020 11:59PM. 

Sincerely,

Adya Misra, PhD

Senior Editor 

PLOS Medicine

plosmedicine.org

Requests from Editors:

Abstract-please add summary demographics for the cohort.

Author summary

Please tone down “We also found that females and the youngest adults received lower absolute benefits of statin treatment when compared to males and the oldest adults” to reflect that this is a finding from a modelling study. I suggest “found” is replaced with “our models suggest”.

The Data statement requires revision, as you say all data are available but then you say “Some data are available from https://www.cdc.gov/nchs/nhanes/index.htm The remaining data cannot be shared publicly because they are from the REGARDS…”. Can you please clarify within the data statement which data can and cannot be accessed freely or otherwise. Please note that authors cannot be the contact persons for data requests and must be deposited with a data committee or an ethics committee or similar. For each data source used in your study: 

The link to REGARDS researchers and institutions in the acknowledgements is broken. Please correct as needed. 

The four guidelines need to be clearly highlighted in the methods as it currently doesn’t stand out 

The data sources are mentioned in suppl methods but brief details in main methods needed

Please can you provide p-values throughout, as needed. Note that we require exact p values, unless p<0.001

Could you replace the word “harm” with an alternative, such as adverse effects or side effects ? The same goes for benefits, perhaps replace with effects?

References should be in Vancouver style please

The STROBE checklist must be called out in the methods and please remove page numbers from the checklist as these are subject to change. Instead please use paragraphs and sections. 

Did your study have a prospective protocol or analysis plan? Please state this (either way) early in the Methods section.

Comments from Reviewers:

Reviewer #1: Authors have addressed most of my comments.

Minor correction

In your Author summary, Page 4, you write: "Projected differences by age and sex also were become more pronounced as the effect of statins on T2D was increased". You may want to delete either "were" or "become".

Thank you.

Reviewer #2: Thanks authors for their great effort to improve the manuscript. All my comments were professionally addressed. I am satisfied with the response and the revision. No further issues needing attention.

[LINK]

---

## [Editor Report · Decision Letter 2]

22 Jul 2020

Dear Dr. Engeda, 

On behalf of my colleagues and the academic editor, Dr. Leopold Aminde , I am delighted to inform you that your manuscript entitled "Projections of incident atherosclerotic cardiovascular disease and incident type 2 diabetes across evolving statin treatment guidelines and recommendations: A modelling study" (PMEDICINE-D-19-04150R2) has been accepted for publication in PLOS Medicine. 

PRODUCTION PROCESS

PRESS

PROFILE INFORMATION

Thank you again for submitting the manuscript to PLOS Medicine. We look forward to publishing it. 

Best wishes, 

Adya Misra, PhD

Senior Editor 

PLOS Medicine

plosmedicine.org